# A Dual Adjuvant System for Intranasal Boosting of Local and Systemic Immunity for Influenza Vaccination

**DOI:** 10.3390/vaccines10101694

**Published:** 2022-10-11

**Authors:** Fumi Sato-Kaneko, Shiyin Yao, Fitzgerald S. Lao, Yukiya Sako, Jasmine Jin, Nikunj M. Shukla, Howard B. Cottam, Michael Chan, Masiel M. Belsuzarri, Dennis A. Carson, Tomoko Hayashi

**Affiliations:** Moores Cancer Center, University of California San Diego, 9500 Gilman Dr, La Jolla, CA 92093-0809, USA

**Keywords:** intranasal pulmonary boosting regimen, influenza virus infection, TLR4 agonist, TLR7 agonist, combination adjuvant, liposomal formulation

## Abstract

Systemically vaccinated individuals against COVID-19 and influenza may continue to support viral replication and shedding in the upper airways, contributing to the spread of infections. Thus, a vaccine regimen that enhances mucosal immunity in the respiratory mucosa is needed to prevent a pandemic. Intranasal/pulmonary (IN) vaccines can promote mucosal immunity by promoting IgA secretion at the infection site. Here, we demonstrate that an intramuscular (IM) priming-IN boosting regimen with an inactivated influenza A virus adjuvanted with the liposomal dual TLR4/7 adjuvant (Fos47) enhances systemic and local/mucosal immunity. The IN boosting with Fos47 (IN-Fos47) enhanced antigen-specific IgA secretion in the upper and lower respiratory tracts compared to the IM boosting with Fos47 (IM-Fos47). The secreted IgA induced by IN-Fos47 was also cross-reactive to multiple influenza virus strains. Antigen-specific tissue-resident memory T cells in the lung were increased after IN boosting with Fos47, indicating that IN-Fos47 established tissue-resident T cells. Furthermore, IN-Fos47 induced systemic cross-reactive IgG antibody titers comparable to those of IM-Fos47. Neither local nor systemic reactogenicity or adverse effects were observed after IN delivery of Fos47. Collectively, these results indicate that the IM/IN regimen with Fos47 is safe and provides both local and systemic anti-influenza immune responses.

## 1. Introduction

Influenza viruses initially infect and replicate on the mucosal surface of the upper and lower respiratory tracts resulting in severe symptoms, especially in vulnerable populations, including young children, the elderly and immunosuppressed populations [1,2,3]. Most distributed influenza vaccines (e.g., inactivated virus and subunit vaccines) are administered systemically through the intramuscular (IM) route and have provided minimal effects on mucosal protection in respiratory tract [4,5,6]. Due to a lack of sufficient local protection, systemically vaccinated individuals may still allow the influenza virus to replicate, shed from mucosal surfaces, and spread infections. In the respiratory tract, mucosal immunoglobulin (Ig) A and resident memory T cells play essential roles in immune defense [4,7,8]. Secretory IgA in the respiratory mucosa inhibits viral entry by blocking virus attachment to target cells [9,10,11,12]. Intranasal and/or pulmonary (IN) vaccination induces functional IgA, as demonstrated by the observation that IgA purified from mice IN vaccinated with a split influenza vaccine can protect naïve recipient mice from virus colonization in the lung after virus challenge [13,14]. Influenza virus infection and IN vaccination using a live-attenuated virus (LAIV) and recombinant subunit vaccines also induce lung tissue-resident T cell responses and protect against lethal virus challenges in mice [4,7,15]. Hence, IN vaccines that induce humoral and cellular immunity in the airway are important to protect individuals from respiratory virus infections.

To improve vaccine protective efficacy, adjuvants are employed to enhance the immunogenicity of co-administered antigens [16,17,18]. Several adjuvants have recently been approved by the Food and Drug Administration (FDA) for vaccines against virus infections [16,19]. For example, MF59, (squalene and surfactants), AS03 (𝛼-tocopherol, squalene, and polysorbate80 in oil-in-water emulsion), AS01B (containing a Toll-like receptor (TLR) 4 agonist monophosphoryl lipid A (MPLA) and a saponin QS-21 in a liposomal formulation), and phosphorothioate oligonucleotide (CpG1018, a TLR9 agonist) are used in vaccines for influenza (MF59, AS03), shingles (AS01B), and hepatitis B (CpG), respectively [16,18,20,21,22,23]. However, these adjuvants were only approved for IM administration and no licensed adjuvant for IN vaccination is currently available in the U.S. In 2000–2001, the detoxified heat-labile enterotoxin of *Escherichia*
*coli* was used in intranasal influenza vaccines. However, this adjuvanted vaccine was withdrawn because of local reactogenicity, including the development of Bell’s palsy in some vaccine recipients [24,25,26]. Thus, safe adjuvants for IN administration that can induce local protection against influenza infection are highly desirable [4,7,27].

Our series of adjuvant discovery studies yielded a combination synthetic adjuvant, designated Fos47, for an influenza vaccine in murine models [28,29,30,31]. Fos47 comprises small molecule TLR4 (2B182C) and TLR7 (1V270) agonists in a liposomal formulation, that is also referred to as Lipo-(1V270+2B182C) in [32]. In a murine model, IM prime-boost vaccination with inactivated influenza virus (IIAV) adjuvanted with Fos47 (IM-Fos47) enhanced germinal center reactions in the draining lymph nodes, induced sustained and balanced systemic Th1/Th2 responses, and provided homologous protection from lethal infections. In addition, IIAV adjuvanted with Fos47 elicited cross-reactive antibody responses against heterologous influenza virus strains [32]. The IM-Fos47 exhibited minimal local and systemic reactogenicity in mice. The minimal local and systemic reactogenicity of Fos47 prompted us to study its potential as a mucosal adjuvant in the respiratory tract. To take advantage of the breadth and duration of protective immune responses induced by the IM-Fos47 vaccination, we immunized mice by IM injection as priming and boosted via the IN route, and evaluated both lung mucosal and systemic immune responses to IIAV.

## 2. Materials and Methods

### 2.1. Mice

Eight-week-old BALB/c mice (female and male) were purchased from the Jackson Laboratory (Bar Harbor, MA, USA). The mice were first acclimated for at least 48 h after arrival at a vivarium, and then used for studies. All mouse experiments were performed at University of California (UC) San Diego Animal Facility. The animal husbandry was performed by UC San Diego Animal Care Program, including 24 h veterinary care support. All animal experiment protocols in this study received prior approval by the Institutional Animal Care and Use Committee (IACUC) for UC San Diego (approved protocol number S00028).

### 2.2. Reagents

Inactivated Influenza A virus (IIAV) A/California/04/2009 (H1N1)pdm09 was obtained from BEI Resources Repository (# NR-49450, Manassas, VA, USA). A TLR7 agonist 1V270 [28] and a TLR4 agonist 2B182C were synthesized in our laboratory as previously described [32]. Fos47 [Lipo-(1V270+2B182C) in reference [32] consists of 20 μM 1V270 and 4 mM 2B182C in a DOPC (1,2-dioleoyl-sn-glycero-3-phosphocholine) liposomal formulation. Fos47 and liposomal vehicle (blank liposomes; Lipo-Veh) were prepared by Inimmune Corp. (Missoula, MT, USA) as previously described [32]. In brief, the DOPC:cholesterol liposomes were produced with a mass ratio of 60:15 mg/mL, respectively. Liposomes were prepared using the thin lipid -film rehydration method. Thin films were rehydrated with 10 mM sodium phosphate at pH 7.1, with compounds (Fos47) or without compounds (Lipo-Veh). The formulations were then sonicated in a bath sonicator. AS01B, a combination vaccine adjuvant of MPLA and saponin in a liposomal formulation, is one of the components of Shingrix (zoster vaccine, manufactured by GlaxoSmithKline Biologicals) [33]. Shingrix was purchased from Henry Schein Inc. (Melville, NY, USA), and only the adjuvant suspension component AS01B (NDC 58160-829-03) was used in this study. RPMI-1640 (#11875), penicillin-streptomycin (#15140-122), PBS (#14190144) and HSBB (#4175095) were purchased from Thermo Fisher Scientific (Waltham, MA, USA). Bovine serum albumin (BSA, #A3059) was obtained from Sigma (St. Louis, MO, USA). Fetal bovine serum (#FB-12) was purchased from Omega scientific (Tarzana, CA, USA).

### 2.3. Immunization with IIAV plus Fos47 through IM and IN Routes

IIAV (10 μg/injection) was mixed with Fos47 (1 nmol/injection 1V270 + 200 nmol/injection 2B182C) or AS01B [10 μL/injection (1/50th of human adult dose)] in 50 μL and was used as a full dose. Lipo-Veh (50 μL/injection) that contained an equivalent number of liposomes without compounds, served as a negative control. AS01B was used as a positive control adjuvant. For IM administration, 50 μL of each agent was injected into the gastrocnemius muscle using a 29G insulin syringe. For IN boosting, mice were anesthetized with 2.5% isoflurane (Fluriso^TM^, VetOne, Boise, ID, USA) in O_2_ (flow rate 1.8 L/min) using a vaporizer (Veterinary anesthesia systems Co., Bend, OR, USA) and a half dose of the vaccine (25 μL) was intranasally delivered slowly dropwise to the nares using a pipetman (Gilson Middleton, WI, USA). Immunization schedules and sample size are indicated in each figure legend.

### 2.4. Bronchoalveolar Lavage and Nasal Lavage for Collecting Mucosal Fluids and Cells

Mice were euthanized using CO_2_, and bronchoalveolar lavage (BAL) procedures were performed immediately thereafter. Cold 1%BSA-PBS (900 μL) was slowly delivered into the lung through the trachea, and then the fluid was withdrawn by gentle suction using a 1 mL syringe with a blunt needle attached (#8881202363, Cardinal Health, Dublin, OH, USA). The BAL fluids (BALFs) were spun, and the supernatants and the cell pellets were collected as BALF and BAL cell samples. For nasal lavage, the lower jaw was removed to expose the nasopharynx. The nasal cavity was gently rinsed through the nasopharynx with 300 μL of 1%BSA-PBS using a 1 mL syringe fitted with a 22G catheter (#SR*OX2225CA, TERUMO, Somerset, NJ, USA) and the flow-through from the nares was collected in a 1.5 mL tube. The fluids were spun to remove debris and the supernatants were assessed as nasal lavage specimens. The nasal lavage samples were stored at −20 °C, while the BAL cells were stored at −80 °C, until further use.

### 2.5. Assessment for IgG and IgA Antibody Titers against Homologous Virus

IgG and IgA against hemagglutinin (HA) and neuraminidase (NA) antigens of the homologous virus were measured by ELISA. Assays were performed using half area 96 well plates (#3690, Corning, Corning, NY, USA) as previously described [34]. Detailed information for reagents and samples is described in Appendix A, respectively. In brief, plates were coated with recombinant HA or NA proteins of A/California/04/2009 (H1N1), blocked, then incubated with serially diluted sera, BALF or nasal lavage samples in blocking buffer. After incubation at 4 °C overnight, plates were washed and detecting antibodies were added. After washing and incubation with *p*-nitrophenyl phosphate substrate (*p*-NPP), plates were read at 405 nm and 570 nm as a reference on a plate reader (Tecan, Switzerland). We prepared standard sera with known endpoint titers as previously reported [32,34]. Briefly, standard sera were obtained from mice hyper-immunized with IIAV [A/California/04/2009 (H1N1), homologous strain]. The endpoint titer of the standard sera was calculated as a reciprocal of the highest dilution that gave an absorbance reading double the absorbance of the background. Each plate had a titration of this standard sera to generate a standard curve. The titer of the test sample was interpolated from the standard serum titration from the same plate. The results are expressed in units per milliliter (arbitrary units).

### 2.6. Evaluation of Cross-Reactive IgG and IgA by ELISA

The levels of cross-reactive IgG and IgA were measured by ELISA as described above with some modifications. Recombinant HA and NA proteins of heterologous virus strains used for coating are presented in Appendix A. Plates were coated with each protein overnight at 4 °C for IgG or at room temperature (RT) for IgA. For total IgG, sera were diluted at 1:50 in blocking buffer followed by 4-fold serial dilutions (Appendix A). Geometric means of total IgG titer curves are presented in the results. For IgA, diluted nasal lavage and BALF specimens were incubated overnight at RT (Appendix A). OD_405–570nm_ values were measured by a Tecan plate reader. The standard sera prepared above were for measurement of antibodies against homologous strain, and thereby are not appropriate to measure cross-reactive antibody titers. Therefore, adjusted optical density (OD)_405–570nm_ values are presented for cross-reactive antibody titers. To assess phylogenetic relationships of HA and NA proteins, the amino acid sequences of the proteins obtained from the Influenza Research Database (https://www.fludb.org/brc/home.spg?decorator=influenza, accessed on 23 March 2022) were aligned by the MUSCLE algorithm (Multiple Sequence Comparison by Log-Expectation) [35]. The phylogenetic tree was constructed by the Neighbor-joining method [36] using Molecular Evolutionary Genetics Analysis software MEGAX (https://www.megasoftware.net/, accessed on 23 March 2022) [37].

### 2.7. Flow Cytometry Analysis of Lung Cells

To enrich for tissue-resident immune cells in the respiratory tract, circulating cells in vaccinated mice were labeled by intravenous injection with anti-CD45-PE/Cy7 antibody (2.4 μg/100 μL PBS/injection) 5 min prior to euthanasia [38]. Immediately after sacrifice, mice were gently perfused with 10 mL HBSS through the right ventricle to remove circulating cells. Harvested lungs were cut into 1–2 mm^3^ pieces and were digested at 37 °C in HBSS containing collagenase I (50 μg/mL; #LS004196, Worthington Biochemicals, Lakewood, NJ, USA) and Deoxyribonuclease I (DNase I) (20 pg/mL; #2138, Worthington Biochemicals) using gentleMACS^TM^ Octo Dissociator (Miltenyi Biotec, Germany). The lung cell suspensions were passed through a 100 μm cell strainer (#250377, Genesee Scientific) to remove debris, followed by washing with 2% FBS-PBS. After hemolysis using ACK lysis buffer (#A10492-01, Thermo Fisher Scientific), cells were washed with PBS and rinsed with 2% FBS-PBS. The isolated lung cells were incubated with anti-mouse CD16/32 antibody for blocking Fc receptor in Stain Buffer (#554657, BD Biosciences, La Jolla, CA, USA), followed by incubation with a cocktail of antibodies plus tetramer reagents. Stained cells were washed with PBS and then incubated with propidium iodide to determine live/dead cells. Detailed information for antibodies and reagents used in flow cytometry analysis and gating strategy is presented in Appendix A and Appendix A, respectively. MHC class I tetramer and class II tetramer reagents bearing the influenza A HA peptides with amino acid sequences IYSTVASSL (533-541) and SFERFEIFPKE (127-137), respectively, were obtained from the NIH tetramer core facility. Data were acquired using MACSQuant^®^ Analyzer 10 (Miltenyi Biotec) and analyzed using FlowJo (version 10.8.1, Becton Dickinson, Ashland, OR, USA).

### 2.8. Assessment of Recall T Cell Responses

Splenocytes isolated from vaccinated mice using gentleMACS^TM^ Octo Dissociator (1 × 10^6^ cells/200 μL/well) were cultured with RPMI supplemented with 10% FBS, 100 units/mL, penicillin and 100 μg/mL streptomycin in the presence of 10 μg/mL HA [A/California/04/2009 (H1N1); #11055-V08B, Sino Biological, Houston, TX, USA]. Three days later, culture supernatants were harvested and stored at −20 °C until further use. IFNγ, IL-5, and IL-17 release in the culture supernatants were evaluated by ELISA. Detailed information for reagents is described in Appendix A.

### 2.9. Safety Studies

BALB/c mice (female and male) were IN administered with 25 μL PBS, Lipo-Veh, Fos47, or AS01B (5 μL AS01B plus 20 μL PBS). BAL sample was collected at indicated periods to obtain BALF and BAL cells. Chemokine/cytokine levels in BALF and sera were quantified using a ProcartaPlex^TM^ Multiplex Immunoassay kit according to the manufacturer’s instructions (Thermo Fisher Scientific). Interferon stimulated gene (ISG) expressions in BAL cells were determined with a Quantigene Plex Assay kit by following the manufacturer’s protocols [39] (Thermo Fisher Scientific). Lung and head/nose samples were collected 24 h after IN administration for histologic analysis. Lungs were filled with 800 μL Formalde-Fresh^®^ (#SF93-4, Fisher Scientific, Hampton, NH, USA), and the tracheas were tied with a string. The lungs were then harvested and fixed with Formalde-Fresh^®^ overnight. Head/nose samples were decalcified and fixed using Cal-Ex^®^ II (#CS511-1D, Fisher Scientific) for 4–5 days. These tissue samples were embedded in paraffin, sliced into 5 μm sections, and stained with hematoxylin and eosin (HE) at the Tissue Technology Shared Resource at UC San Diego. Histology images were obtained by a microscope BZ-X800 using 2× and 20× objective lenses (KEYENCE, Itasca, IL, USA). For mobility assessment, mouse behavior was monitored for 3 h post administration, and body weights were measured daily for 7 days. For complete blood counts (CBC), whole blood was collected in a microtainer tube coated with K_2_EDTA (#365974, BD, Franklin Lakes, NJ, USA). CBC and serum chemistry analyses were performed by the UC San Diego Animal Care Program Diagnostic Services Laboratory, using the instruments HemaVet Hematology and VetScan2(VS2) Biochemistry, respectively.

### 2.10. Statistical Analyses

Data from in vivo studies are expressed as means with standard error of mean (SEM). To compare multiple groups with continuous/ordinal outcomes, Kruskal–Wallis tests with Dunn’s post hoc test were applied. To compare two groups (e.g., IN-Fos47 vs. IM-Fos47), the two-tailed Mann–Whitney test was used (antibody titers and immune cell populations). For time course analysis (chemokine/cytokine levels and ISG expressions) two-way ANOVA was applied where time was used a factor variable, and the Holm–Sidak method was used for multiple comparison adjustments. Repeated measures using two-way ANOVA with Holm–Sidak post hoc tests were applied to compare days 56 and 182 in each group in antibody kinetics data, and to compare to a Lipo-Veh-group in body weight data, with the Geisser and Greenhouse epsilon hat method to account for unequal variances in outcome values across groups. Prism 9 software (GraphPad Software, San Diego, CA, USA) was used. P values smaller than 0.05 were considered statistically significant.

## 3. Results

### 3.1. Enhancement of Local and Systemic Antigen-Specific IgA Titers by IN Boosting with Fos47

In respiratory infectious diseases such as influenza, secretory IgA in the lung mucosa can block virus attachment and entry into host cells [4,7,11]. Vaccination with LAIV through IN route induces high secretory IgA titers in the upper and lower respiratory tracts and provides protection from virus infections [4,7]. Hence, we tested whether an IN boosting regimen with Fos47 could enhance local antigen-specific IgA responses as a biomarker for protection. BALB/c mice were IM primed with inactivated influenza virus [IIAV; A/California/04/2009 (H1N1)pdm09] adjuvanted with Fos47 on day 0. Liposomal vehicle (Lipo-Veh) that was prepared without 1V270 and 2B182C served as a negative control (Figure 1A). At 21 and 28 days thereafter, the mice were boosted with a half dose of the same agents via IM (herein referred to as IM-Lipo-Veh or IM-Fos47) or IN route (herein referred to as IN-Lipo-Veh or IN-Fos47). A half dose of the vaccine (25 μL) was administered twice, one week apart, for boosting immunization, due to the limitation of IN dosing volume in mice. AS01B, a licensed liposomal combination adjuvant with a TLR4 agonist MPLA and a saponin, served as a positive control [33]. Seven days after the last boosting (on day 35), sera and BALF were collected. The titers of IgA specific to HA of A/California/04/2009 (H1N1) in serum and BALF were evaluated by ELISA.

We initially evaluated local and systemic IgA antibody titers (BALF and serum, respectively) (Figure 1A–C) in the following comparisons: (1) IN-Fos47 and IN-Lipo-Veh, (2) IM-Fos47 and IN-Fos47, and (3) IN-Fos47 and IN boosting with AS01B (the approved adjuvant; IN-AS01B). The IN-Fos47 booster induced significantly higher anti-HA(H1, Cal09) IgA titers in serum and BALF samples compared to IN-Lipo-Veh (** *p* < 0.01, Figure 1B,C). When IM-Fos47 and IN-Fos47 were compared, IN boosting with Fos47 enhanced IgA antibody titers in both serum and BALF (^††^  *p* < 0.01, Figure 1B,C). IM-Fos47 induced only a baseline level of IgA specific for HA(H1, Cal09) in both BALF and sera (Figure 1B,C). Notably, the IN-Fos47 promoted IgA secretion at comparable levels to that of IN-AS01B (ns in Figure 1B,C).

To further validate the effects of IN boosting with Fos47 on IgA secretion in the upper respiratory tract, mice were immunized with the IM-IN-IN regimen and nasal lavage specimens were collected on day 35 (Figure 1D). The IgA titers in nasal lavage specimens were enhanced by IN-Fos47 compared to IN-Lipo-Veh (*p* < 0.001, Figure 1E).

### 3.2. Mucosal IgA Induced by IN Boosting with Fos47 Is Cross-Reactive

To assess cross-reactivity, we evaluated the binding of IgA in the nasal lavage and BALF to recombinant HA proteins of A/Puerto Rico/8/1934 (PR8, H1N1), A/Victoria/3/1975(H3N2), and A/Netherland/219/2003 (H7N7). The phylogenetic relationship of these HA proteins is presented in Figure 2A, including the strain used for vaccination [A/California/04/2009(H1N1pdm)]. A/Puerto Rico/8/1934 (PR8, H1N1), A/Victoria/3/1975(H3N2), and A/Netherland/219/2003 (H7N7) are antigenically distinct from A/California/04/2009. Among these three heterologous strains, PR8 (mouse adapted H1N1 variant strain) belongs to group 1 and is phylogenetically closer to A/California/04/2009, while A/Victoria/3/1975 (H3N2) and A/Netherland/219/2003 (H7N7) belong to group 2 [40]. In the nasal lavage, IgA induced by IN-Fos47 was cross-reactive to variant HA(H1 PR8) (Figure 2B). Nasal lavage IgA also displayed a trend of cross-reactivities to HA proteins from H3 and H7 (Appendix A). IN-Fos47-induced mucosal IgA in BALF conferred wider cross-reactivity to variants HA(H1, PR8), HA(H3) and HA(H7) (Figure 2C–E). Together, the IN boosting with Fos47 induced significantly higher cross-reactive local IgA compared to IN-Lipo-Veh.

### 3.3. IN Boosting with Fos47 Enhanced Local and Systemic Antigen-Specific T Cell Responses

To optimally block respiratory viral infections, both humoral immunity and cellular immunity play an important role [41,42,43]. In response to infection or vaccinations, activated memory T cells persists at mucosal sites. These tissue-resident memory T cells contribute to mucosal protective immunity [44,45]. Tissue-resident memory T cells in the lung mucosa have been reported to mediate site-specific viral clearance and protection from a lethal viral challenge [46,47,48]. To assess whether the IN-Fos47 generates tissue-specific T cell immunity in the lung mucosa, lymphocytes were isolated from the lungs of IN-boosted mice on day 35 (1 week after the last boosting) and analyzed by flow cytometry. We employed an intravascular (iv) antibody labeling technique using an anti-CD45 antibody to distinguish circulating cells (ivCD45^+^) from non-circulating tissue-resident cells in the lung (ivCD45^–^, tissue-resident) [38]. The ivCD45^–^ cells were further stained for CD69, a canonical tissue-resident memory T cell marker [44]. The percentages of CD69 expressing CD44^+^CD4^+^ and CD44^+^CD8^+^ memory T cells in the gated ivCD45^–^CD4^+^ and ivCD45^–^CD8^+^ cell populations, respectively, in the lungs of Fos47 IN-boosted mice were significantly higher than those of IN-Lipo-Veh-boosted mice (*p* < 0.01, Figure 3A,B and Appendix A and Appendix A). To assess whether the lung tissue-resident memory T cell responses were specific for influenza HA, we assessed by flow cytometry the binding of class I and class II MHC tetramer reagents, which detect T cell receptors (TCRs) specific to HA peptides with amino acid sequences IYSTVASSL and SFERFEIFPKE, respectively. The proportions of MHC class II tetramer^+^ cells in the gated resident (ivCD45^–^) CD4^+^ memory T cells were significantly increased in the Fos47 IN boosted mice compared to the IN-Lipo-Veh control mice. (*p* < 0.001, Figure 3C). The percentages of MHC I tetramer^+^ cells in the gated resident CD8^+^ memory T cells were also increased by IN boosting with Fos47 compared to IN-Lipo-Veh (*p* < 0.001, Figure 3D). These data suggest that the IN boosting with Fos47 may enhance the establishment of mucosal cellular immunity in situ.

To evaluate systemic HA-specific T cell responses after the IN-Fos47 boosting regimen, we measured antigen (HA)-specific IFNγ, IL-5 and IL-17 release by splenocytes following restimulation with HA ex vivo [49]. IFNγ, IL-5, and IL-17 in the culture supernatants were significantly enhanced in the IN-Fos47 group compared to the IN-Lipo-Veh group (*p* < 0.05, Figure 3E). These results imply that the IN-Fos47 enhances both local and systemic antigen-specific T cell responses.

### 3.4. Induction of Systemic IgG by IN Boost with Fos47 Is Comparable to That Induced by Original IM-Boosting Regimen

We have previously demonstrated that the IM-prime and IM-boost with IIAV adjuvanted with Fos47 (original IM-Fos47 regimen) induced influenza virus-specific IgG responses sufficient to protect mice from live virus challenge [32]. Systemic IgG induced by IIAV with the original IM-Fos47 regimen reacted not only to HA of homologous strains but also with heterologous HA and NA [32]. Thus, we next asked whether IN-Fos47 regimen could induce high titer, long-lasting, and broadly reactive systemic IgG equivalent to the original IM-Fos47.

BALB/c mice were primed with IIAV adjuvanted with Fos47 and boosted via IM or IN routes (Figure 4A) [32]. One week after the last boosting (days 28 and 35 for IM and IN boosting regimens, respectively), the serum IgG2a and IgG1, which are Th1 and Th2 response-mediated isotypes, respectively, were measured by ELISA [50]. The IN-Fos47 induced HA(H1, Cal09)-specific IgG2a equivalent to the original IM-Fos47 (Figure 4B). Both IN- and IM-Fos47 moderately increased anti-HA(H1, Cal09) IgG1 production (Figure 4C). Following the IN-Fos47 boosting regimen, HA-specific IgG2a and G1 persisted on days 56 through 182 (Figure 4D–G).

To examine further the breadth of IgG responses, we determined the cross-reactivity of IgG1 and IgG2a induced by IN-Fos47 to HA and NA of homologous and heterologous influenza virus strains, A/California/04/2009 (H1N1) and A/Netherland/219/2003 (H7N7), respectively [40] (Figure 2A, (Figure 5A and Appendix A). IN-Fos47-induced IgG2a and IgG1 were also highly reactive to homologous NA (Figure 5B,C). Furthermore, IN-Fos47 induced significantly higher IgG against H7 and N7 compared to IN-Lipo-Veh (Figure 5D,E). Taken together, the above data indicate that IN-Fos47 yielded long-lasting and cross-reactive systemic IgG responses against influenza antigens, equivalent to the original IM-Fos47.

### 3.5. IN Administration with Fos47 Did Not Cause Local and Systemic Reactogenicity

Case–control studies have indicated that IN administration of vaccines increases the risk of facial nerve paralysis, (Bell’s palsy) [24]. This side effect may be due to local inflammatory responses induced by antigen, vehicle or added adjuvant. Thus, it was important to assess whether IN administration of Fos47 may cause any local and systemic inflammation. To this end, we measured chemokine and pro-inflammatory cytokine release in BALF and serum at 4, 6 and 24 h following the administration of Fos47 or AS01B. In BALF, IN delivery of Fos47 minimally elevated IL-6, TNFα and KC at 4 h, but by 6 h the values had returned to the baseline (IN-administered Lipo-Veh) (Figure 6A). IN administration with Fos47 also demonstrated minimal effects on systemic cytokines (Figure 6B). Of note, IN administration of AS01B induced significantly higher inflammatory cytokines levels which peaked at 6 h (Figure 6A,B).

Because excessive type I interferon (IFN) release often causes leukocytopenia, we also tested whether Fos47 induced the expression of type I IFN stimulated genes (ISG15 and ISG56) by Quantigene plex assay. Time-course gene expression analyses at 2 to 48 h post-administration demonstrated that IN administration with Fos47 did not enhance ISG15 or ISG56 expression, whereas AS01B transiently enhanced the expression of the genes that peaked at 6 or 24 h (Figure 6C). Histologic analysis of nostril and lung sections revealed no inflammatory cells, exudates, and cellular detritus in any groups 24 h post IN administration (Appendix A). We also monitored both body weights and behavioral changes following IN-administration of Lipo-Veh, Fos47, and AS01B. No significant changes were observed in mice IN administered with Fos47 (Figure 6D). Consistent with the cytokine and behavioral data, complete blood counts and serum chemistry profiles after IN administration with Fos47 remained unchanged (Appendix A). These results suggest that the IN-Fos47 boosting regimen is non-inflammatory, well tolerated, and devoid of systemic and local adverse effects.

## 4. Discussion

Most currently licensed vaccines against respiratory pathogens (e.g., influenza virus and SARS-CoV-2) are administered via IM route and confer systemic antibody protection specific to the strains contained in the immunogen [4,7]. However, circulating antibodies induced by IM vaccines may provide incomplete local protection and thereby cannot fully prevent viral infection and shedding from the respiratory tracts [4,7,8]. Strong local antiviral immunity is required to control the spread of contagious respiratory virus infections. We previously demonstrated that the IM vaccination of IIAV adjuvanted with Fos47 (original IM-Fos47) enhanced influenza-specific IgG production and protected mice from live viral challenges [32]. Systemic anti-viral IgG induced by IM vaccination prevents severe disease and death due to the infection [51]. In contrast, IN route of IIAV vaccination induces less systemic anti-viral IgG, while it produces higher mucosal IgA compared to IM-vaccination [52]. In this study, to confer both mucosal and systemic protective anti-viral immune responses, we employed IM priming in combination with IN boosting with Fos47. A combination of IM priming and IN boosting with Fos47 induced local antigen-specific IgA (Figure 1) and lung tissue-resident memory T cells (Figure 3). Importantly, the mice that were IN boosted with Fos47 produced serum HA-specific IgG2a and IgG1 comparable to those in mice that were vaccinated with the original IM-Fos47 regimen. Collectively, the IN-Fos47 boosting regimen was thus sufficient to augment both local and systemic anti-influenza immune responses.

IN vaccines for respiratory viral infections are known to enhance mucosal antiviral immunity, involving both local antigen-specific antibody secretion and the accumulation of antigen-specific tissue-resident T cells [53,54,55]. Since respiratory mucosal tissues are the first line of defense against microbial infections, they are also equipped with a tight network of innate immune cells, e.g., dendritic cells and NK cells [11,45,56]. The IN vaccination route is also attractive because it does not use needles and hence does not require trained healthcare personnel for administration [55,57]. Despite these considerable advantages, only a limited number of IN vaccines (e.g., FluMist Quadrivalent and Fluenz Tetra; the nasal spray of LAIV) against respiratory viral infection are available for relatively younger adults [45]. Furthermore, only a few vaccine adjuvants for mucosal administration have been tested in humans. Enterotoxins, e.g., cholera toxin (CT) and *Escherichia coli* heat-labile toxin (LT), are well-known mucosal adjuvants. However, these enterotoxins induced severe diarrhea in humans due to ADP-ribosyltransferase activity and are unsuitable for clinical use [58]. Relatively non-toxic (detoxified) variants of CT and LT have been developed and tested in humans with IIAV. However, the trial was halted because some individuals developed facial nerve palsy (Bell’s palsy) [24,25,59]. Thus, a safe and non-inflammatory vaccine adjuvant for mucosal administration is highly desired. Our experiments demonstrated that IN administration of Fos47 transiently induced low levels of chemokines/cytokines without increasing expression of IFN downstream genes in bronchial cells. Consistent with these results, IN administration of the adjuvant did not influence body weights, CBC, or serum chemistry parameters (Figure 6). These results imply that Fos47 is a safe and promising vaccine adjuvant for mucosal administration.

Natural infection with influenza virus and immunization with recombinant virus proteins induce secretory IgA and lung tissue-resident memory B cells expressing various Ig isotypes, including IgA [53,54]. Enhanced secretory IgA at the infection site correlates with higher virus neutralization, and with both homologous and cross-protection against lethal virus challenges [9,10,11,13,14,60]. Further studies have indicated that secreted IgA plays critical roles both in protection against homologous IAV infection and in cross-protection against nasal infection by viral variants and elevated IgA levels have been correlated with protective efficacy [54,61,62,63,64]. In this study, secreted IgA titers were used as a biomarker for protective efficacy.

Local protective immunity against respiratory virus infection involves multiple immune components, especially, antigen-specific secretory IgA and lung tissue-resident T cells [27,55]. A recent report described that IN boosting with a SARS-CoV-2 adenovirus-vectored vaccine following IM priming induced local mucosal IgA and resident T cells in the lungs, which were responsible for protection from the subsequent virus challenge [65]. The IN-Fos47 yielded antigen-specific IgA cross-reactive against multiple influenza virus strains in the nasal lavage and BALF (Figure 1). Furthermore, we detected HA-specific lung tissue-resident memory T cells following IN-Fos47 (Figure 3). These findings imply that the IN-Fos47 would provide local protection against virus challenges.

For IN delivery, 10 to 50 μL per dose have been commonly administered in mouse models [66,67]. In this study, the IM dose (50 µL) was split into two doses to lower the volume to 25 µL for IN delivery. A volume greater than 5 μL spreads across the upper (nose) to lower respiratory tracts (lung) [66]. Therefore, we investigated the IN regimen as intranasal/pulmonary vaccination, and adverse effects on the nose, lung and circulatory system were assessed.

Although the IN-Fos47 could establish strong local antiviral immune responses, it is essential to confirm that the IN-Fos47 also induced robust and sustained systemic immunity. Thus, we evaluated serum antibody responses induced by IN-Fos47 according to these criteria: (1) the titers of antigen-specific IgG2a and IgG1, (2) the durability of IgG production, and (3) the cross-reactivity of IgG antibodies to different strains of influenza virus. As a comparison for systemic antibody responses, we employed the original IM regimen with Fos47, which was previously proven to induce anti-influenza immune responses sufficient for protection against virus challenge [32]. Serological analysis of IgG in sera one week after the last boost demonstrated that the IN-Fos47 adjuvant regimen induced equivalent IgG2a titers to IM-Fos47 boosting, which persisted for 6 months (Figure 4). Finally, the IN-Fos47 induced IgG cross-reacted with HA and NA of heterologous virus strains, similar to responses induced by IM-Fos47 [32] (Figure 5). Since IN vaccination can be self-administered without a visit to a health provider, additional boosting with IIAV and Fos47 via IN route may enhance the potency and durability of vaccine protection in human populations.

## 5. Conclusions

In summary, immunization with IIAV adjuvanted with Fos47 using an IM priming–IN boosting regimen induced both local and systemic anti-influenza antibody responses and established lung tissue-resident memory T cells, which are hallmarks for protective immune responses against influenza virus challenge. IN administration of Fos47 was neither reactogenic nor inflammatory, making it suitable for repeated administration. Thus, we believe that the IN-Fos47 has the potential to serve as an effective IN adjuvant for human respiratory virus infections.

## Figures and Tables

**Figure 1 vaccines-10-01694-f001:**
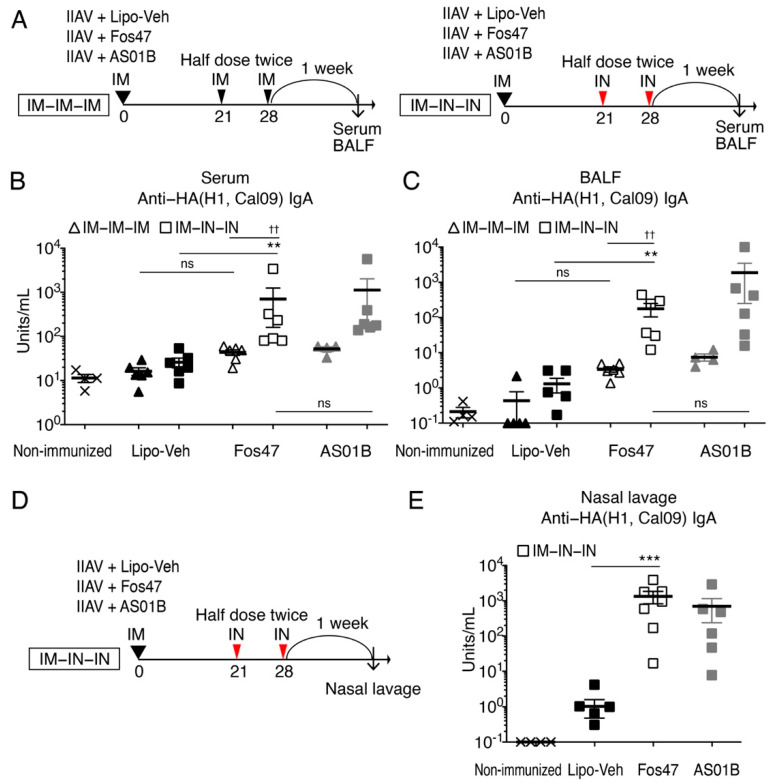
IgA antibody titers in serum and BALF were enhanced by IN boosting with Fos47. (**A**) Experimental protocol. Female BALB/c mice (*n* = 6/group) were IM primed with IIAV plus Lipo-Veh, Fos47 or AS01B on day 0. The mice were either IM or IN boosted with a half dose of the same vaccine on days 21 and 28. Lipo-Veh and AS01B served as negative and positive controls, respectively. The mice were sacrificed on day 35 and BALF and serum were collected. Non-immunized mice served as a negative control. (**B**) HA(H1, Cal09)-specific serum IgA. (**C**) HA(H1, Cal09)-specific BALF IgA. Bars indicate means ± SEM. ** *p <* 0.01, ns; there was no significant difference, Kruskal–Wallis with Dunn’s post hoc test among 4 groups (IM-Lipo-Veh, IM-Fos47, IN Lipo-Veh and IN-Fos47). ^††^
*p <* 0.01, ns; there was no significant difference, two-tailed Mann–Whitney *U*-test to compare two groups (IM-Fos47 vs. IN-Fos47 or IN-Fos47 vs. IN-AS01B). (**D**) Female BALB/c mice (*n* = 6–7/group) were IM primed and IN boosted with IIAV plus Lipo-Veh, Fos47 or AS01B. Nasal lavage specimens were collected on day 35 to test IgA levels. (**E**) Anti-HA(H1, Cal09) IgA antibody titers in nasal lavage. Bars indicate means ± SEM. *** *p <* 0.001, two-tailed Mann–Whitney *U*-test to compare two groups (IN-Lipo-Veh vs. IN-Fos47).

**Figure 2 vaccines-10-01694-f002:**
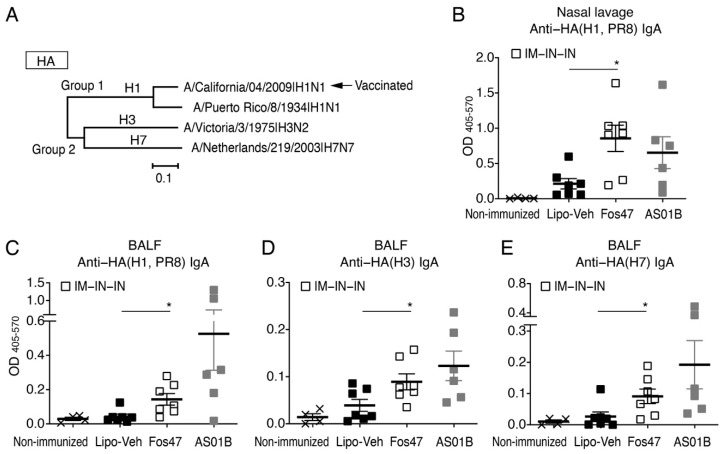
Cross-reactive IgA was induced by IN boosting with Fos47. (**A**) Phylogenetic analysis of influenza virus strains used in the cross-reactivity study. Amino acid sequences of HA proteins used in ELISA were aligned by the MUSCLE algorithm using the Influenza Research Database. Phylogenetic trees were constructed by the neighbor-joining method using MEGAX software. A/California/04/2009(H1N1pdm09) was used for vaccination. Cross-reactivity against antigenically distinct heterologous virus strains, A/Puerto Rico/8/1934(H1N1), A/Victoria/3/1975(H3N2) and A/Netherlands/219/2003(H7N7), were tested in this assay. (**B**–**E**) Female BALB/c (*n* = 6–7/group) mice were vaccinated with IIAV plus Lipo-Veh, Fos47 or AS01B by IM-IN-IN regimen. Nasal lavage and BALF were collected on day 35. Non-immunized mice served as a negative control. IgA binding to recombinant HA protein of A/Puerto Rico/8/1934(H1N1), A/Victoria/3/1975(H3N2), or A/Netherlands/219/2003(H7N7) were evaluated by ELISA. (**B**) anti-HA (H1, PR8) IgA in nasal lavage. (**C**) Anti-HA(H1, PR8) IgA in BALF. (**D**) Anti-HA(H3) IgA in BALF. (**E**) Anti-HA(H7) IgA in BALF. Bars indicate means ± SEM. * *p <* 0.05, two-tailed Mann–Whitney *U*-test to compare IN-Lipo-Veh and IN-Fos47.

**Figure 3 vaccines-10-01694-f003:**
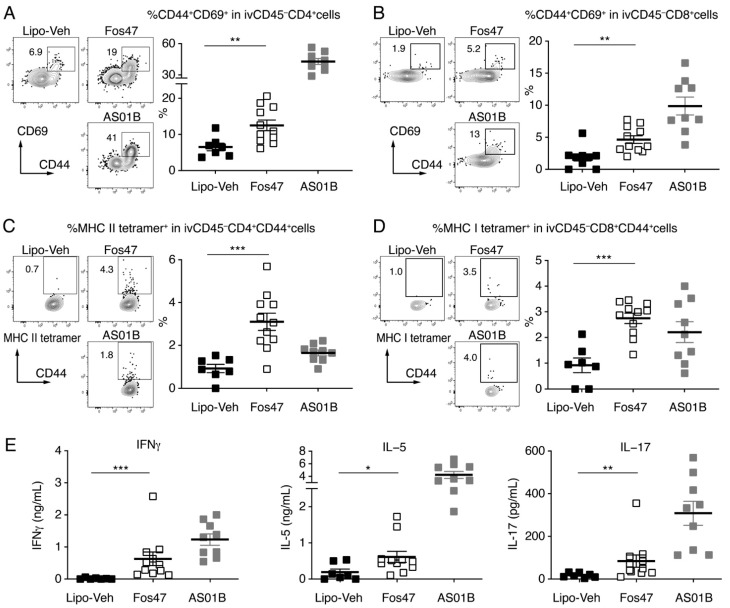
IN boosting with Fos47 enhanced antigen-specific T cell responses in the lung. Female BALB/c mice (*n* = 7–11/group) were immunized with IIAV adjuvanted with Lipo-Veh, Fos47 or AS01B by IM-IN-IN regimen on days 0, 21 and 28, and on day 35 lung cells were isolated and analyzed by flow cytometry. Non-circulating tissue-resident cells in the lung were identified by iv staining with anti-CD45 antibody. Non-circulating (ivCD45^–^) CD4^+^ and CD8^+^ cells were analyzed for CD44^+^CD69^+^ memory T cells. (**A**) %CD69 expressing CD44^+^ in ivCD45^–^CD4^+^ and (**B**) ivCD45^–^CD8^+^ T cells are presented. (**C**,**D**) ivCD45^–^ CD4^+^CD44^+^ (**C**) and ivCD45^–^CD8^+^CD44^+^ (**D**) memory T cell subsets were further analyzed using MHC class II tetramer and MHC class I tetramer reagents by flow cytometry. (**E**) Splenocytes were isolated on day 35 and restimulated ex vivo with 10 μg/mL HA [A/California/04/2009(H1N1)] protein for 3 days. IFN γ, IL-5 and IL-17 secreted in the supernatants were evaluated by ELISA. Pooled data from two independent experiments are presented. Bars indicate means ± SEM. *****
*p* < 0.05, ** *p* < 0.01, *** *p* < 0.001, two-tailed Mann–Whitney *U*-test to compare IN-Lipo-Veh and IN-Fos47.

**Figure 4 vaccines-10-01694-f004:**
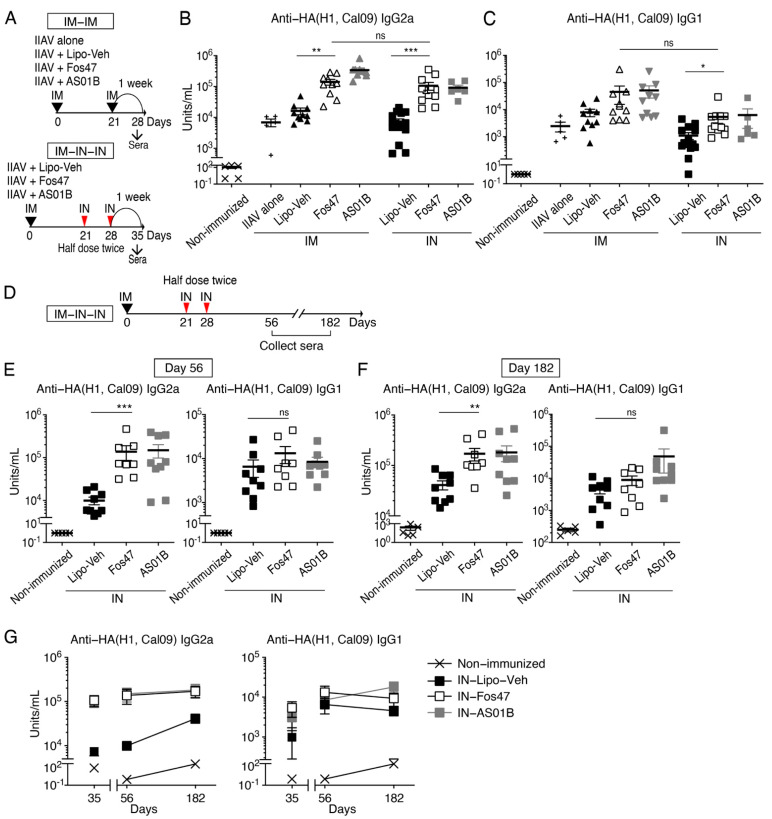
Antigen-specific IgG antibody titers induced by IN boosting with Fos47 were comparable to IM boosting with Fos47. (**A**) Experimental protocols of IM-IM and IM-IN-IN regimens. BALB/c mice were IM primed with IIAV plus Fos47 or Lipo- Veh (blank liposomes, negative control) in 50 μL on day 0. For IM-IM regimen, the mice were IM boosted on day 21 with a full dose (50 μL per dose). For IM-IN-IN regimen, the mice were IN administered with a half dose of the vaccine (25 μL per dose) twice 7 days apart. Non-immunized (naïve) and IIAV alone served as negative controls. One week after the last boost, sera were collected to evaluate serum IgG2a and IgG1 against HA. (**B**) Anti-HA(H1, Cal09) IgG2a. (**C**) Anti-HA(H1, Cal09) IgG1. Bars indicate means ± SEM (*n* = 6–12 group). * *p <* 0.05, ** *p <* 0.01, *** *p <* 0.001, ns; not significant, Kruskal–Wallis with Dunn’s post hoc test to compare 4 groups (IM-Lipo-Veh, IM-Fos47, IN-Lipo-Veh and IN-Fos47). (**D**–**F**) Long-term serological analysis of IM-IN-IN regimen. Female BALB/c mice (*n* = 6–12/group) were IM primed on day 0 and IN boosted on days 21 and 28. HA(H1, Cal09)-specific IgG2a and IgG1 in sera were evaluated on days 56 (**E**) and 182 (**F**) ** *p* < 0.01, *** *p* < 0.001, ns; there was no significant difference, Mann–Whitney *U*-test to compare IN-Lipo-Veh and IN-Fos47. (**G**) Time course plots of antibody titers. The data for individual mice for days 35, 56 and 182 are presented in Figure 4B,C,E,F, respectively. Bars indicate means ± SEM (*n* = 6–12 group). There was no significant difference between days 56 and 182 in each group by repeated measures two-way ANOVA with Holm–Sidak’s post hoc test.

**Figure 5 vaccines-10-01694-f005:**
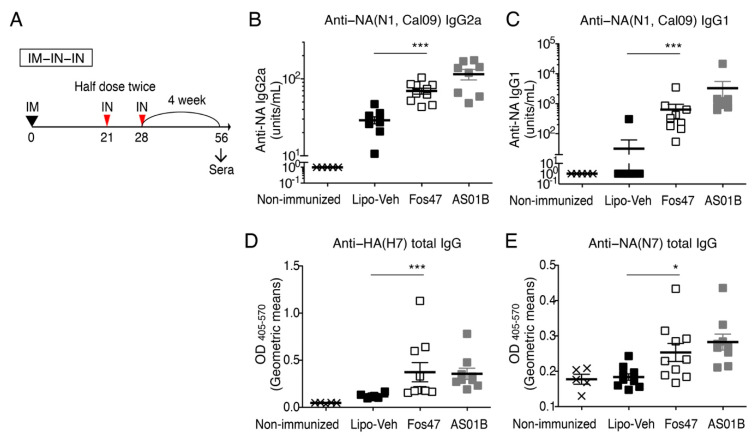
IM-IN-IN combination regimen with Fos47 enhanced serum IgG responses against NA and phylogenetically distinct virus strains. Female BALB/c mice (*n* = 5–10/group) were IM primed with IIAV plus Lipo-Veh, Fos47 or AS01B. The mice were IN boosted with a half dose of the same agents on days 21 and 28 (IM-IN-IN). Sera were collected on day 56. (**A**) Experimental protocol. IgG2a (**B**) and IgG1 (**C**) specific to NA [A/California/04/2009 (H1N1), homologous strain] were evaluated by ELISA. Units/mL (arbitrary units) are presented. (**D**,**E**) Total IgG cross-reactive to H7 (**D**) and N7 proteins (**E**) of A/Netherlands/219/2003(H7N7) were measured by ELISA. Phylogenetic relationships of HA and NA proteins are depicted in Figure 2A (for HA) and Appendix A (for NA). Geometric means of OD are presented. Bars indicate means ± SEM. * *p <* 0.05, *** *p <* 0.001, two-tailed Mann–Whitney *U*-test to compare IN-Lipo-Veh and Fos47.

**Figure 6 vaccines-10-01694-f006:**
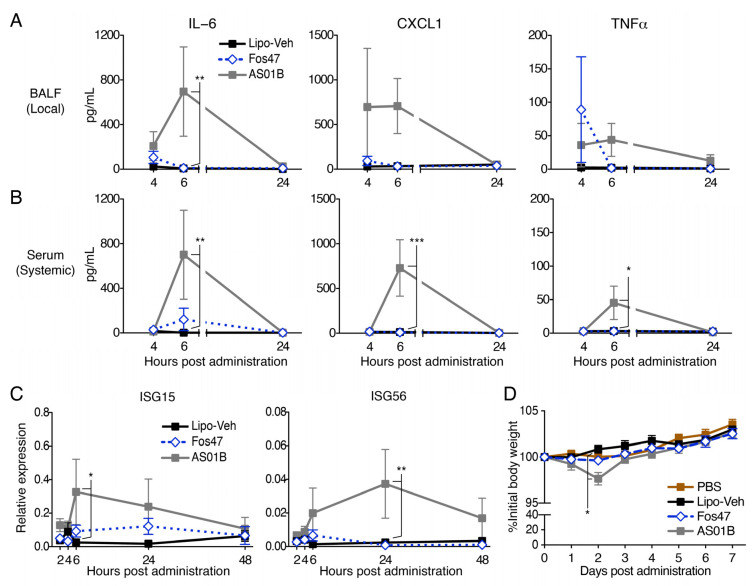
Intranasal delivery of Fos47 demonstrated minimal local and systemic cytokine and chemokine induction. (**A,B**) BALB/c mice (female *n* = 2–3 and male *n* = 2–3, total *n* = 4–6) were IN administered with 25 μL Lipo-Veh, Fos47 or AS01B. IL-6, CXCL1 and TNFα in BALF (**A**) and serum (**B**) at 4, 6, 24 h were evaluated by Multiplex cytokine assay. (**C**) ISG15 and ISG56 expression in BAL cells were examined at indicated time points (BALB/c mice; female *n* = 3 and male *n* = 2–3, total *n* = 5–6/group). The gene expressions were evaluated by the Quantigene assay and were normalized to the house keeping gene expression of Rpl19 and Rps20. (**D**) Body weight changes after IN administration. Female BALB/c mice (*n* = 4/group) were IN administered with 25 μL PBS, Lipo-Veh, Fos47 or AS01B. %Initial body weights for 7 days are presented. Bars indicate means ± SEM. * *p <* 0.05, ** *p <* 0.01, *** *p <* 0.001, two-way ANOVA with Holm–Sidak’s post hoc test (A-D: vs. Lipo-Veh).

## Data Availability

All data are available in the main text or Appendix A.

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
