# Peer review of "A Dual Adjuvant System for Intranasal Boosting of Local and Systemic Immunity for Influenza Vaccination"

_vaccines, 2022, doi:10.3390/vaccines10101694_

Round 1

Reviewer 1 Report

Sato-Kaneko et al. evaluate here a liposomal dual TLR4/7 adjuvant (Fos47) as adjuvant for vaccination with an inactivated influenza A vaccine and compare homologous intramuscular with heterologous intramuscular/intranasal vaccination.

·         Methods: Please provide more detail on the animal experiments according to ARRIVE guidelines, e.g. sex of the mice

·         Methods: Please ass the dose for Lipo-Veh

·         Line 250-252: It is not clear to what Fos47 is compared and to what it is superior. This holds true also at other parts of the results section, e.g. line 296.

·         Are all comparisons not marked in the figures non-significant?

·         Line 372: Please change the two PMIDs to correctly formatted references

·         Figure 1B: Why is no data for the im-im-im group shown?

·         Figure 1 and also other experiments: Where also non-immunized mice as proper negative control used? Please add.

·         Figure 1: Why are two different statistical tests used for comparisons within the same graph.

·         Figure 2: Data on the nasal lavage is not shown for all strains. Please add if available.

·         Figure 2: It would be helpful to calculate antibody titers, e.g. as endpoint titers instead of showing OD.

·         Figure 4A: Is there a rational that the second im is in this experiment not split to two doses as the intranasal dose and the second im dose in the previous experiments?

·         Figure 4 E + F could be replaced by Figure S2 to show individual values in the main figures. I believe this adds information. In the last line of the legend “S3” should be replaced by “S2”.

Reviewer 2 Report

This paper by Sato-Kaneko et al. describes the use of a dual TLR4/TLR7 adjuvant system that enhances mucosal and systemic immunity following intranasal (IN) delivery when compared to intramuscular (IM) administration using an influenza vaccination model. Authors reported enhanced antigen-specific IgA secretion and tissue resident memory T cells following IM priming-IN boosting compared to IM priming/boosting regimes. Regarding this I have a couple of comments:

1. Are these mucosal antigen-specific IgA  and systemic IgG antibodies able to neutralize viral entry? It would be good if authors can test whether these antibodies can confer protection from infection?

2. Why the authors tested an IM priming-IN boosting regime and they did not evaluate a double IN priming/boosting regime instead? The latest might have given better responses/protection. 
